# Effects of Epiphytes and Depth on Seagrass Spectral Profiles: Case Study of Gulf St. Vincent, South Australia

**DOI:** 10.3390/ijerph16152701

**Published:** 2019-07-29

**Authors:** Charnsmorn Hwang, Chih-Hua Chang, Michael Burch, Milena Fernandes, Tim Kildea

**Affiliations:** 1Department of Environmental Engineering, National Cheng Kung Chung University, Tainan City 701, Taiwan; 2Global Water Quality Research Center, National Cheng Kung University, Tainan City 701, Taiwan; 3Department of Ecology & Evolutionary Biology, The University of Adelaide, Adelaide, South Australia 5005, Australia; 4Australian Water Quality Centre, SA Water, Adelaide, South Australia 5000, Australia; 5College of Science and Engineering, Flinders University, Adelaide, South Australia 5001, Australia

**Keywords:** seagrass, reflectance, epiphytes, growing depth, optically shallow coastal waters, remote sensing, benthic bottom type

## Abstract

Seagrasses are a crucial indicator species of coastal marine ecosystems that provide substratum, shelter, and food for epiphytic algae, invertebrates, and fishes. More accurate mapping of seagrasses is essential for their survival as a long-lasting natural resource. Before reflectance spectra could properly be used as remote sensing endmembers, factors that may obscure the detection of reflectance signals must be assessed. The objectives in this study are to determine the influence of (1) epiphytes, (2) water depth, and (3) seagrass genus on the detection of reflectance spectral signals. The results show that epiphytes significantly dampen bottom-type reflectance throughout most of the visible light spectrum, excluding 670–679 nm; the depth does influence reflectance, with the detection of deeper seagrasses being easier, and as the depth increases, only *Heterozostera* increase in the exact “red edge” wavelength at which there is a rapid change in the near-infrared (NIR) spectrum. These findings helped improve the detection of seagrass endmembers during remote sensing, thereby helping protect the natural resource of seagrasses.

## 1. Introduction

Seagrasses are underwater angiosperms (flowering plants) that tend to form dense beds in shallow, subtidal marine and estuarine benthic seafloors that could be optically clear [1,2,3]. The development and maintained existence of these seagrass beds as a natural resource helps create a substratum and provide shelter, and food sources for epiphytic algae, mollusks, invertebrates, fishes, and sea turtles [4,5,6,7,8]. This, in turn, also allows for the trapping and stabilization of sediments, which helps in maintaining water clarity [1,9,10,11,12]. Because of these effects, seagrasses contribute to the quality of coastal marine ecosystems by improving the biodiversity and productivity within these environments [1,3,13,14,15,16,17,18]. However, although seagrasses are crucial to coastal habitats worldwide, they are in decline, with 29% of the known seagrass areal extent having been lost since 1879 [12,19]. In particular, coastal South Australia has experienced a 21% seagrass reduction [20], with its metropolitan coastal waters undergoing an average annual loss of 85 ha yr^−1^ until the late 2000s [21,22], with a slow come back in the last decade [23,24,25]. These historical declines emphasize the need for more accurate mapping of seagrasses in the area.

Spectral imaging remote sensing is a proven, useful tool for the detection of seagrass distribution, particularly due to advances in spaceborne and airborne detection sensors [2,26,27,28,29,30]. In particular, since remote sensing spectral reflectances vary along the electromagnetic light spectrum, distinct optical signatures may be repeatedly measured [2,28,31,32,33] with fewer limitations due to cost, time, and inaccessibility [34,35]. Underwater benthic communities, however, are only detected in the visible light spectrum (≈350–750 nm), where light may penetrate the water column without it being readily absorbed and reflected back to a sensor [14,27,36]. Additionally, derivative analyses of hyperspectral data have been used in a number of studies to determine inconspicuous peak features. Although there has been a debate between the use of first-order derivatives [9,37,38,39] compared to second- and fourth-order derivatives [40,41,42,43], the use of these derivatives helps minimize the number of key bandwidths to analyze. In doing so, derivative analyses provide a method to isolate statistically significant spectral differences between benthic bottom types. While baseline knowledge and continual monitoring of seagrass distribution are increasingly critical for mitigation of any further losses, relatively little is known about the factors that may obscure the detection of the reflectance signals such as epiphytes and varying natural growing depths [34,38].

Aquatic epibionts are organisms such as algae, bacteria, sponges, crustaceans, and mollusks that grow on the surface of seagrasses [44,45,46,47]. Note, although the term epiphyte has been defined variously throughout literature [44,45,47,48,49], this paper follows previously extensively reviewed protocol and define an epiphyte as a plant-based epibiont using a plant as a substrate [44]. Nutrient inputs such as nitrate and ammonium promote epiphyte, plankton, and macroalgae growth; thereby, potentially reducing the light that is available for the photosynthetic surfaces of seagrass plants, as well as increasing the drag [50,51]. In general, higher loads of epiphytes are considered detrimental to seagrasses since they reduce growth, cause thinning of shoots, reduce above-ground biomass, and ultimately cause seagrass death [50]. Seagrass complexes (i.e., seagrass and associated epiphytes), however, have been noted to uptake nitrogen from the water column [21]. *Posidonia* seagrass complexes account for 49–99% of the total biological uptake of nitrate [21], but *Amphibolis* seagrass complexes have higher uptake rates of ammonium than compared to *Posidonia* [51]. Seagrasses and their associated epiphytes were estimated to uptake about one-third of all ammonium discharged to the Adelaide region in the mid-2000s, but less than 1% of the nitrate [21].

Generally, the more structurally complex the seagrass, the greater number of epibiont assemblages and a greater degree of epibiont biomass they contain. There is substantial variation in the number of epiphytic algae based on seagrass species [45]. The spatial distribution of epiphytes on seagrass leaves is influenced by the leaf morphology and growth rate [48,52], although the specific side of the leaf, water flow, and canopy height are known to influence epiphyte distribution as well [52]. Due to the basal growth pattern for the strap-shaped leaves of *Posidonia* [45], the observed trend of epiphyte cover for this seagrass occurs from the base to the tips of the leaves, where the tips of the leaves are older and consequently have a larger amount of time to accumulate epiphytes [51]. For example, 70 taxa of epiphytes have been found in the leaves of *Posidonia* consisting of diatoms, cyanobacteria, dinoflagellates, red algae, and green algae [53]. Similarly, within the canopies of *Amphibolis* meadows, epiphytes tended to accumulate in significantly higher biomasses at the tips of the splayed straps of leaf blades [45]. The leaves of certain *Amphibolis* support larger biomasses of epiphytes comprised of algae and epifaunal grazers than those compared to *Posidonia* [53]. Conversely, the much smaller size and simpler structure of *Zostera* and *Heterozostera* seagrasses have no marked difference in epiphyte abundance based on the spatial distribution of seagrass leaves [45]. However, the effect of epiphytic algae on *Heterozostera* can vary greatly, with epiphytes contributing from 8% to 49% of the total above-ground biomass in these systems ([45,54], respectively).

Although previous research has been done on epiphytes, there have been limited studies that directly studied the measured changes in the natural growing depth among multiple seagrass genera. Instead, many of these studies primarily focus either on the light availability at extreme depths as a limiting factor, often noting canopy growth [52,55,56,57] or controlled systems with specific incremental depths [9,58]. For example, in a study limited to only *Posidonia* seagrass, the leaf width and leaf length showed little variability between shallow and deep sites, although there were differences in frond weight and leaf area per shoot [59]. Similarly, in another study of only one benthic type, reflectance responses of the aquatic plant known as *Ceratophyllum demersum* were measured in a manipulated experiment using a submersible panel and a large, 20,000 L water tank [58]. The results from that study provided a useful reference to the “700 peak”, which is also called the “red edge” due to the rapid change in reflectance within the near-infrared at approximately this waveband.

There is a clear need to understand the influence of epiphytes and varying natural growing depths on the spectral signatures of seagrasses. Specifically, while epiphytes have been shown to affect reflectance responses, the influence of specific seagrass genera is still unclear. In addition to this, there are a limited number of clear and thorough studies of spectral signals from multiple seagrass genera collected from natural growing depths. Therefore, the objectives of this paper are to assess the influence of (1) epiphytes; (2) growing depths, and (3) particular genera on the detection of spectral reflectance signatures of seagrasses. 

## 2. Materials and Methods 

### 2.1. Study Area

The study area is separated into two study sites, Bolivar and Seacliff, which are located north and south along the coasts of metropolitan-Adelaide, South Australia, in the optically shallow waters of Gulf St. Vincent, respectively (Figure 1). The Bolivar research site (34°42′55.82″S, 138°26′45.57″E) is located about 1 km northwest of the outfall of the Bolivar Wastewater Treatment Plant, where *Posidonia* is the dominant seagrass, followed by a less dense cover of *Heterozostera* [43,60]. Despite seagrass loss in the area [50], the distribution of these two genera in the northern site has remained consistent for almost a decade [24]. The Seacliff research site (35°2′12.05″ S, 138°30′37.19″ E) is located approximately 0.5 km west of Seacliff Beach, and is dominated by algae and *Amphibolis* seagrass [60,61]. *Posidonia* followed by *Amphibolis* are the dominant seagrass genera in this area, commonly existing as monotypic meadows [62]. *Posidonia* distribution is more fragmented compared to the northern region [62] due to higher wave energy related to the southern region [63].

### 2.2. Field Data Collection

Reflectance profiles for the primary submerged aquatic vegetative communities—*Posidonia*, *Amphibolis*, and *Heterozostera*—were collected roughly every 15 days, weather permitting, at the Bolivar and Seacliff study sites during the summer months (Dec. 2015–Jan. 2016). Limiting conditions for accurate measurements for the calculation of reflectance included: taking measurements only when clouds do not obscure the sensor; fractional cloud cover is less than 20%; the least incident shadow from the boat and sensors are being produced; sea state is calm and does not have significant wave height or whitecap fraction; and sea state does not have dominant swell [64,65]. Multiple sites were sampled within these two locations (Figure 1). The specimens were collected by snorkel brought onto the boat. The specimens were selected from areas known to grow largely in homogeneous (monospecific) patches for each particular seagrass. Submerged benthic bottom types were collected at their natural growing depth, and there was no manipulation of depth increments. Not all bottom types were found naturally occurring at all six growing depths investigated (1.1, 1.5, 2.1, 2.4, 2.8, 3.1 m). Once collected, the specimens were sorted by seagrass genera. In-situ downwelling underwater irradiance (E_d_^−^
_in-situ_) was measured using a cosine corrector when each time latitude/longitude coordinates were changed.

During the collection of the submerged aquatic vegetation, it was apparent that some specimens were ridden with epiphytes. So when epiphytes were obviously present, seagrass specimens were cut in half soon after they were collected in order to separate the leaf blades from the leaf sheaths. The leaf blades, which are found toward the top of the plant, and the leaf sheaths, which are attached to the rhizomes and roots of the plant, generally represent where epiphytes most frequently and least frequently are found on the seagrass plant, respectively. Uneven sample sizes for each seagrass genera were measured for reflectance from both the leaf blades and leaf sheaths.

### 2.3. Measuring Spectral Reflectance

Seagrass specimens in the field were then taken to an outdoor laboratory managed by the Australian Water Quality Center to measure for reflectance spectra on cloud-free days within 3 h of solar noon, wherever possible. An open-air 12 L hydraulic vessel was filled with marine water to a depth of 10 cm. After the interior of the vessel was lined in black [41] to minimize wall effects of the light field [26,41,66], specimens were then placed in the vessel. The reflectance for each submerged seagrass sample was determined using a JAZ-2 spectroradiometer [67] sensor. The sensor, which contains a collecting tip with a fiber optic cable (3 m long, 400 μm diameter) for transmission of light, was attached to a CC-3-UV-S cosine corrector [68] and was also mounted to a non-reflective black-colored pole. The cosine corrector is an optical diffuser which couples to the fiber optic cable and spectroradiometer to collect the signal from a field of view that is 180°. Cosine correctors are typically specified for applications requiring the redistribution of the incident light, such as measuring spectral irradiance of a plane surface in air, water, or other media. 

Using the mounted pole, the sensor, with its attached fiber optic cable and cosine corrector, was placed just beneath the water surface, and pointed at an angle of ~45° nadir from the specimen sample to record the radiance (≈190–890 nm) in the form of upwelling underwater radiance of a particular benthic bottom type (E_u_^−^
_benthic_) and its related upwelling underwater baseline radiance (E_u_^−^
_base_). More than 200 spectra were averaged for each representative specimen with an integration time of 10 microseconds; signal-to-noise ratio (SNR) was reported at 250:1 [69].

Reflectance *R* was calculated using the following equation.
$$R\;\mathrm{benthic}=\;\frac{\left(E_{u^{-}}\;\mathrm{benthic}\right)\;-\;\left(E_{u^{-}}\;\mathrm{base}\right)}{E_{d^{-}}\;\mathrm{in}-\mathrm{situ}}$$
where three variables were measured: upwelling underwater radiance of a specific benthic bottom type (E_u_^−^
_benthic_); upwelling underwater radiance of a baseline, in this case as a black-lined vessel (E_u_^−^
_base_); and on-site downwelling underwater irradiance (E_d_^−^
_in situ_). Data were computed from relative to absolute reflectance by using a Spectralon reference panel; then the spectral database was processed for statistical analyses.

### 2.4. Statistical Analyses

Before processing could occur, the spectral database at each wavelength was tested for normality. Specifically, data points (∑N > 125,000) were tested with the Shapiro–Wilk test and the Kolmogorov–Smirnov test (*p* ≥ 0.10, in both cases) for each nanometer. If necessary, data were then normalized; to maintain homoscedasticity, any data points beyond two standard deviations of each one-nanometer increment’s respective median were considered outliers and subsequently removed from further analyses. In most cases, multiple one-way ANOVAs (analyses of variance) were used to statistically analyze the reflectance spectra in order to address the direct effects of epiphyte presence, natural growing depths, and seagrass genera on spectral signatures without producing an unintended interaction factor. *Posidonia*, *Amphibolis*, and *Heterozostera* were found to be naturally growing at 6, 2, and 3 particular depths, respectively; therefore, ANOVA correlation coefficients of first-, second-, and fourth-order derivative reflectance profiles for *Amphibolis* and *Heterozostera* were not performed. All analyses were performed using R version 3.2.3 (R Foundation for Statistical Computing, Vienna, Austria).

## 3. Results and Discussion

### 3.1. Epiphytes and the Reflectance Signal

In general, reflectance signals from seagrass leaves were significantly dampened by epiphytes throughout the visible light spectrum, except for one particular bandwidth at 670–679 nm (Figure 2 and Appendix A). The reflectance profiles from the leaf blades of seagrasses, where epiphytes are most frequently attached to the leaf surface, were shown to have significantly lower values than those from the leaf sheath (*p* < 0.005). This effect also significantly influenced the reflectance values at 566 nm and 600 nm (*p* < 0.001 for both), thereby potentially weakening the value of the 566:600 band ratio that is used to separate seagrasses from detritus and algae [28,38,70,71]. Epiphytes are known to mask the reflectance of aquatic plant surfaces, which causes altered varying degrees of chl-*a* absorbance [14]. By competing for the absorbance of photons, especially within the red and blue regions [49], epiphytes thereby contribute their own absorption and reflectance features to the spectral response.

In addition to this, epiphytes, which are inherently chlorophyll-based epibionts, may have a more substantial influence on the reflectance response of seagrasses than constituents within marine water. The results in this study show significant dampening effects of epiphytes attaching directly to the seagrass surface at the leaf blades. At the same time, other studies have shown that measuring the reflectance in filtered and unfiltered marine water has no significant differences throughout most of the visible wavelength spectrum [70,71], suggesting that materials in the water column―including phytoplankton, sediment, minerals, and dissolved organic substances [35,55,72,73]—have less impact on the detection of seagrasses than compared to epiphyte presence on the leaf surface. In spite of the strong dampening influence on the detection of seagrasses, the results in this study show that there is one particular bandwidth at 670–679 nm that disregards the effects of epiphytes. 

Beyond this, the 670–679 nm bandwidth discovered in this study can be doubly beneficial for future assessments of seagrasses. Previous research found eleven bandwidths (417, 456, 474, 491, 522, 590, 605, 621, 631, 649, and 681 nm) useful for the differentiation between *Posidonia*, *Amphibolis*, and *Heterozostera* seagrasses [70,71]. Therefore, because this bandwidth falls in the range of one of these eleven bandwidths, isolating and evaluating the reflectance data at approximately 680 nm in a 5 nm increment could be pragmatic and efficient for two reasons. First, first-derivative reflectance could be calculated for the differentiation between seagrasses and other benthic bottom substrates. Second, this bandwidth can still be beneficial to study the differences between seagrasses genera, disregarding the influence of epiphytes. This bandwidth may particularly be useful for future research involving seasonal variations of epiphyte loads where winter months likely allow for less production time of overall chlorophyll, although research in this study involved analyzing spectral reflectance profiles collected during only the summer months.

### 3.2. Reflectance Response due to Natural Growing Depth

Differences in natural growing depths had a significant influence on seagrass spectral response (*p* < 0.004), with two major trends. First, the reflectance tended to decrease as the depth increased between 1.0–2.0 and 2.0–3.0 m, but in deeper ranges of water depth from 3.0–4.0 m, the reflectance actually increased (Figure 3). Additionally, the correlation of depth to reflectance was also calculated when testing the analyses of variance for original, first-order, second-order, and fourth-order spectral profiles. The original (zero-derivative) spectral profiles generally showed nearly zero correlation from 400–650 nm, and then steadily became increasingly moderately negative correlated (*r* = −0.55) from 685–730 nm (Figure 4). The first-derivative correlation to growing depth showed two major troughs where three local minima points showed even stronger negative relationships. Specifically, correlation coefficients reached values as low as −0.75, −0.85, and −0.79 at 516, 598, and 702 nm, respectively (Figure 4). The second-order and fourth-order derivative reflectance correlation values were scattered throughout the visible light spectrum, showing no trend whatsoever.

These findings show that the correlation values of first-derivatives with natural growing depth were more strongly negatively correlated than with the zero-derivative profiles, which was similar to other studies [9,58]. Highly correlated features, negative or positive, have been known to help identify [20,37,39,43,74] and potentially eliminate redundant and numerous feature selections for hyperspectral image classification [30,70,71,75], although correlation itself has been noted to help identify only associations rather than causation [9,58,76,77]. Derivative analysis merely finds and sharpens details in spectral curves that are too subtle to notice and hidden within wide zero-order spectrum bands, thereby highlighting features that were already present in direct spectrophotometry [41,78]. Two of these subtle features, in particular, are that 72% and 62% of the variance at 598 and 702 nm are explained by differences in depth (*r*^2^ = −0.85^2^ and −0.79^2^), respectively. As a consequence, the results in this study suggest that although the 566:600 and 566:689 band ratios has been shown to be an effective tool for the separation of seagrasses and sand from submerged benthic substrates [35,38,70,71], respectively, these ratios are potentially sensitive to fluctuations in seagrass growing depth. Future research may help identify an alternative feature selection for remote sensing of hyperspectral imagery.

Additionally, the results from the F-value calculations suggest that seagrasses growing in shallower depths are more difficult to detect. Within deeper waters of 2–3 m and 3–4 m, the variances in the reflectance measured from 400–750 nm are sufficiently diverse in range, indicating a greater level of confidence (*p* < 0.01) for frequent detection and likely differentiation of seagrasses within these deeper growing depths (Figure 5a). Conversely, the F-values of the reflectance profiles of seagrasses growing at 1–2 m show inconsistent bandwidths throughout the visible light spectrum, where there are sufficient levels of confidence (*p* < 0.05) that they are less easily detected (Figure 5b). These results suggest that reflectance profiles of different vegetative benthic communities may be more difficult to separate in shallower waters. 

Detection of different vegetative benthic communities within optically shallow waters must allow for the sea bottom to be visible, where the influence of upwelling underwater radiance signal from seagrass substra can be measured [79]. However varying levels of chlorophyll, suspended matter, and color dissolved organic matter (CDOM) as well as higher wave and wind energy may cause turbid water environments [2,79,80], where the seagrass beds may not be optically clear. Natural growing depth may also be an alternative factor influencing optical clarity [79]. These causes may lead to misclassification of mapping areas with low cover of seagrass, particularly if utilizing a semi-automated [81] or image pre-processing approach [2].

Although the detection of seagrasses in shallower depths may be more difficult, out of eleven previously identified key bandwidths, five can be used with greater confidence in both shallow and deep waters. In a previous study, eleven bandwidths were considered helpful for differentiating between *Posidonia*, *Amphibolis*, and *Heterozostera* seagrasses [70,71]. Five of these key bandwidths—at 417, 456, 522, 590, and 649 nm—fall in the range of bandwidths that have higher levels of confidence for detection (*p* < 0.01, *p* < 0.01, *p* < 0.01, *p* < 0.01, and *p* < 0.05, respectively). 

### 3.3. Epiphyte “Preference” by Seagrass Genus

Although the presence of epiphytes has a dampening effect for each of the seagrasses measured in this study, particular seagrass genera also contribute a significant role in influencing the spectral responses. Specifically, the leaf sheaths at the bottom plant sections of *Posidonia* and *Amphibolis* had higher reflectance values than compared to the values at the leaf blades (both *p* < 0.001); conversely, the reflectance of the leaf blades for *Heterozostera* had higher reflectance than compared to that of its leaf sheath (*p* < 0.004) (Figure 6). These differences in spectral responses may be attributed to leaf morphology [48,52] and leaf orientation [28,38,43,49]. Seagrasses containing more flattened leaf morphology tend to have no change in epiphyte community composition, whereas those with curved leaves or leaf bundles supported more diverse epiphyte communities on the concave side [48], potentially creating such distinct microhabitats that few species of epiphytes are common to both stem and leaf [45]. Therefore, results within this study suggest that the epiphytes growing on the wide, long, strap-like leaf blades *Posidonia* and on the splayed, fan-like leaf blades of *Amphibolis* had greater advantages for survival and biodiversity than on the shorter, grass-like blades of *Heterozostera* leaves (Figure 7). 

### 3.4. Influence of Natural Growing Depth on Seagrass Spectral Profiles by Genus 

Seagrass genera help to clarify the two major trends observed by the influence of depth on spectral responses. Correlation of original (zero-derivative) spectral profiles and growing depth showed that *Posidonia* followed a similar trend as previously reported for all seagrasses, with an increasingly negative correlation at around 700 nm (Figure 8). *Amphibolis* seagrass followed a slightly increased decline in the negative correlation at 700 nm. However, the correlation of *Heterozostera* increased instead as a positive slope, approaching zero correlation at 700 nm. *Posidonia* and *Amphibolis* had negative correlation slopes of −9.1474 × 10^−3^ and −2.1065 × 10^−3^ from 675–725 nm, while *Heterozostera* instead produced a positive slope of 3.4332 × 10^−3^ within the same bandwidth (Figure 8).

To better understand the phenomena occurring among different seagrass genera, the “700 peak” for each seagrass genera was also assessed at varying water depths. Similar to other findings [9,58], the results in this paper reveal a seagrass genus-specific shift in the exact wavelength at which there is a rapid change of the NIR spectrum. This “700 peak” has been noted to shift to shorter wavelengths in waters that are not optically clear [36,37,38,58,73]. *Posidonia* and, to a lesser extent, *Amphibolis* seem to follow the trend that as the natural growing depth of a seagrass increases, the specific wavelength of this shift decreases (Figure 9). This may be attributed to their plasticity and ability to adjust. *Posidonia* has been known to decouple their light absorption mechanisms to survive the short-term light reduction in deeper water depths due to their larger bodies capable of storing higher quantity and quality of carbohydrates [52]. 

*Heterozostera*, however, seemed to show an opposite trend, where the exact wavelength of the 700 peak instead increased as depth increased (Figure 9). This suggests that the 700 peaks for *Heterozostera* actually shift further in the NIR, potentially making it more difficult for remote sensing detection of submerged aquatic vegetation due to its requirement of being within the visible light spectrum. The results from this study, therefore, show that while *Posidonia* and *Amphibolis* are the dominant seagrasses throughout this region, there is increased difficulty for remote sensing of areas where *Heterozostera* is notably prevalent, such as at the Bolivar Site.

## 4. Conclusions

In this study, the factors affecting bandwidths for optimal seagrass detection—such as natural growing depth, epiphyte location on the plant, and the potential influence of particular seagrass genera—were assessed. Results in this study reveal the following three major findings. First, while epiphytes tend to dampen reflectance signals throughout the visible light spectrum, the 670–679 bandwidth is not affected by these influences, making it useful for improved detection of submerged aquatic benthic communities. Studying this bandwidth may potentially prove especially useful for future research involving distinguishing nuanced differences based on seasonal variations. Second, natural growing depth influences the spectral responses of benthic bottom types, making the detection of seagrasses growing within 1–2 m more difficult than the detection of those growing in deeper water depths of 2–3 and 3–4 m. However, five particular bandwidths known to differentiate between *Posidonia*, *Amphibolis*, and *Heterozostera*—417, 456, 522, 590, and 649 nm—could be used with higher levels of confidence for the detection of seagrasses in shallower and deeper waters. Third, the genus of a particular seagrass plays a role in influencing the “700 peak”, that is, where there is a rapid change of the NIR spectrum. The detection of *Heterozostera* in deeper waters may be more difficult since remote sensing of underwater communities can only be detected within the visible light spectrum. The findings in this study may be used to improve detection of benthic community substrates during remote sensing and monitoring of seagrass distribution. In doing so, submerged aquatic vegetation within coastal environments can be managed more thoroughly and more frequently, thereby mitigating potential anthropogenic impacts and protecting seagrasses as a natural resource. 

## Figures and Tables

**Figure 1 ijerph-16-02701-f001:**
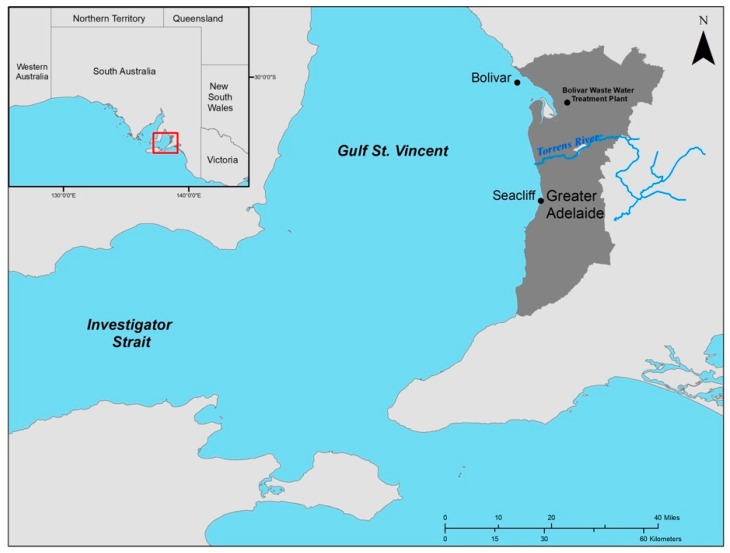
Sampling sites, Bolivar and Seacliff Sites, are north and south of metropolitan-Adelaide in Gulf St. Vincent, South Australia, with specific coordinates as 34°42′55.82″ S, 138°26′45.57″ E and 35°2′12.05″ S, 138°30′37.19″ E, respectively.

**Figure 2 ijerph-16-02701-f002:**
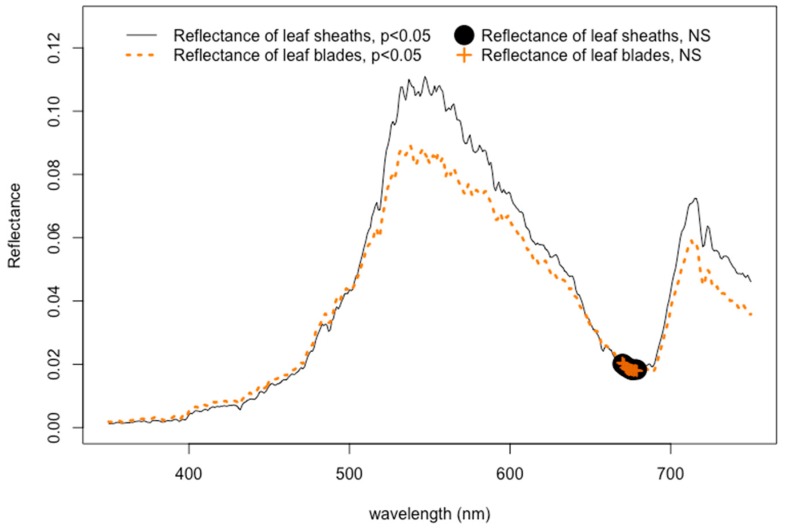
Spectral reflectance profiles for leaf blades and leaf sheaths of seagrass plants. Epiphytes are generally located on leaf blades found at the top of the plants.

**Figure 3 ijerph-16-02701-f003:**
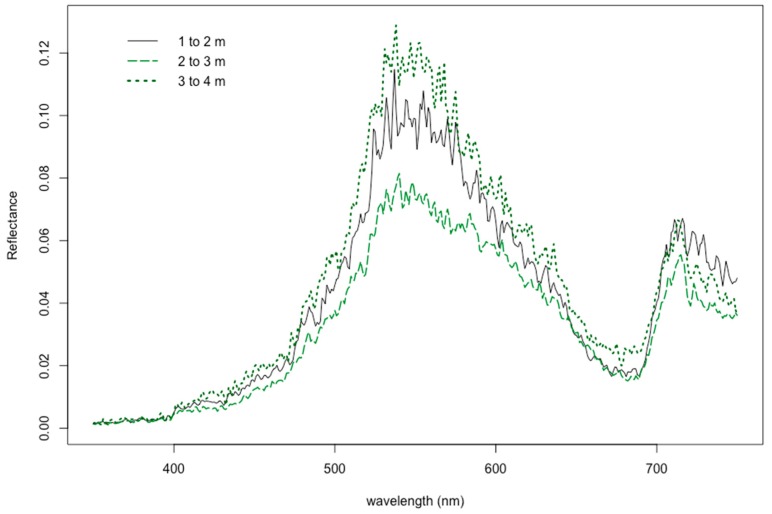
Differences in reflectance response based on natural growing depth.

**Figure 4 ijerph-16-02701-f004:**
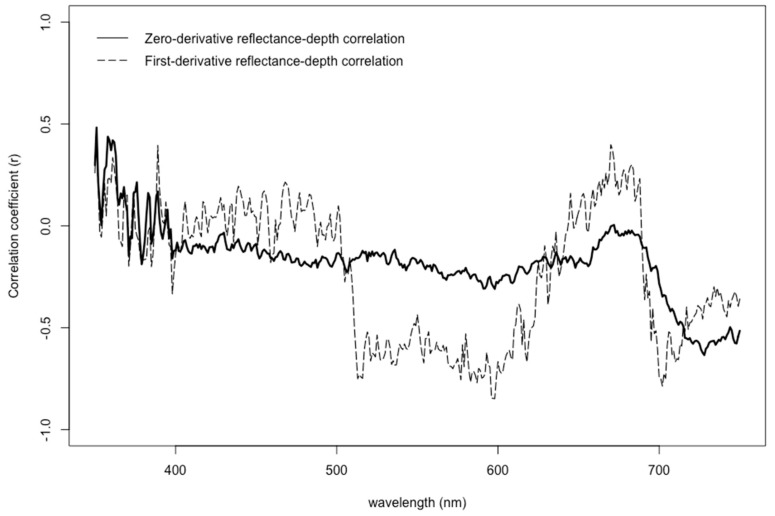
Correlation coefficients between natural growing depths of seagrasses and different derivative reflectance profiles.

**Figure 5 ijerph-16-02701-f005:**
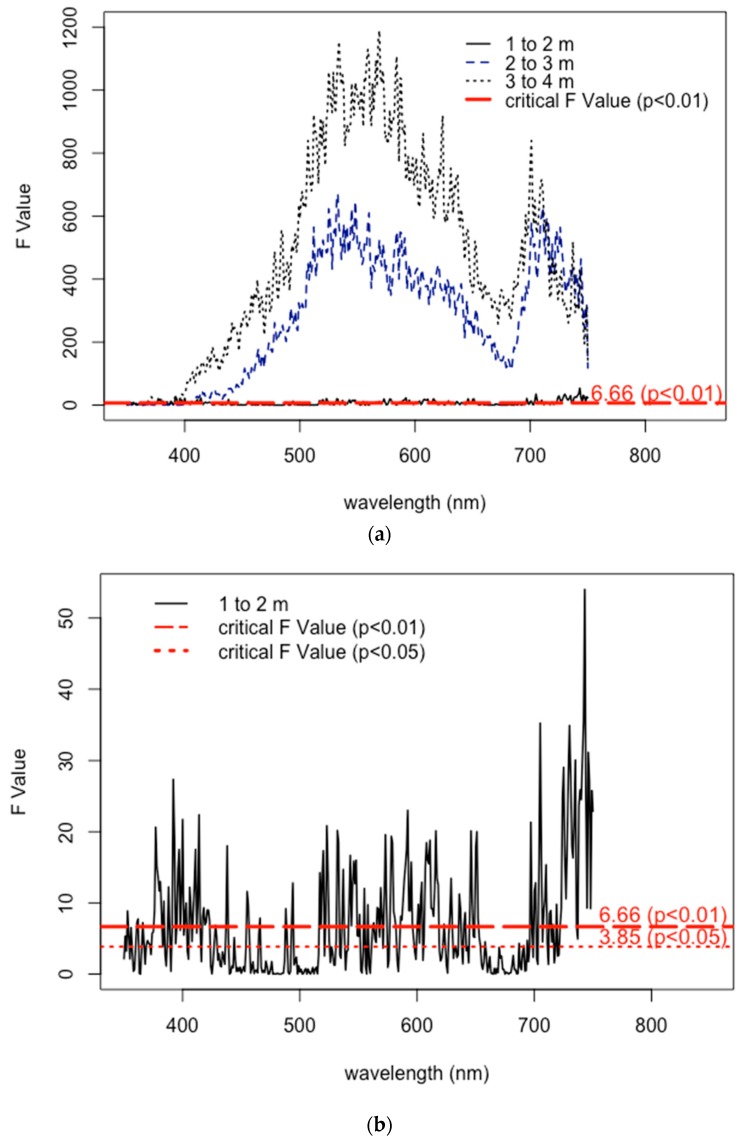
F-values of ANOVA for (**a**) varying natural growing depths; and (**b**) insets for a growing depth of 1–2 m.

**Figure 6 ijerph-16-02701-f006:**
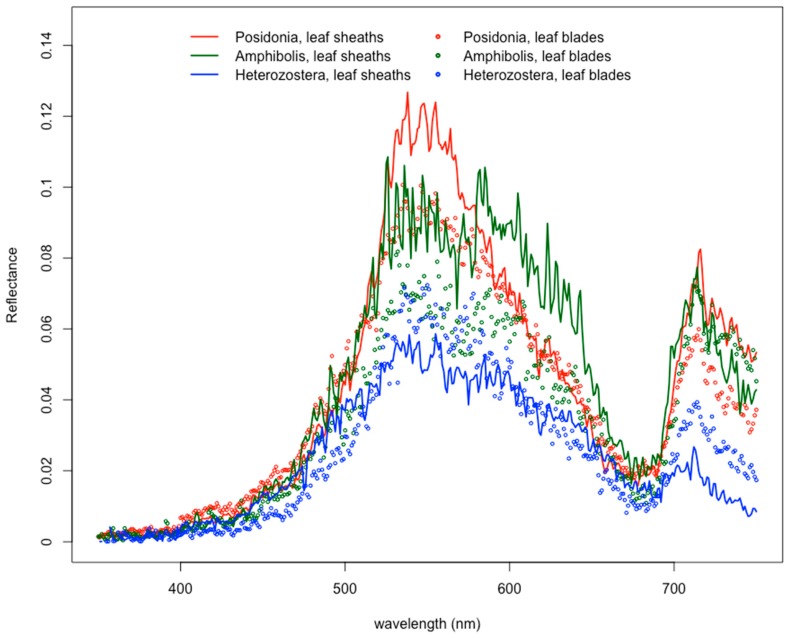
Differences in reflectance spectra based on location of epiphytes on seagrass leaf blades and leaf sheaths.

**Figure 7 ijerph-16-02701-f007:**
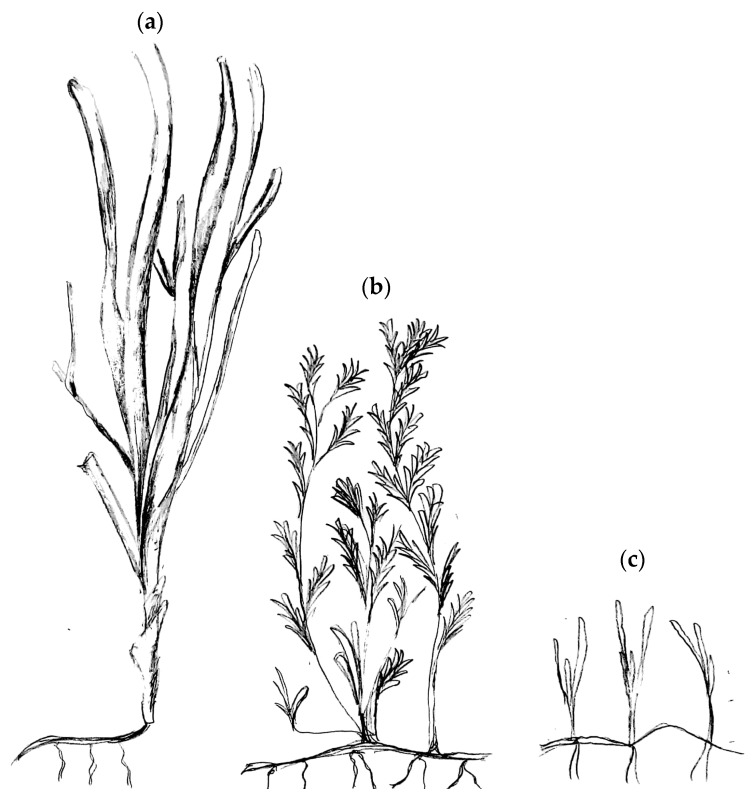
Schematic of seagrass leaf morphology for relative differences in sizes of (**a**) *Posidonia*, (**b**) *Amphibolis*, and (**c**) *Heterozostera* drawn by C. Hwang based on [82]. Note the differences in leaf sheath formation near the roots of each seagrass.

**Figure 8 ijerph-16-02701-f008:**
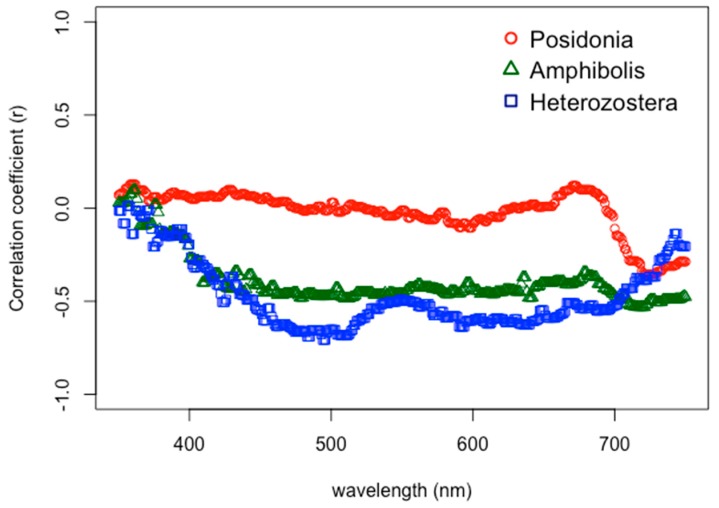
Correlation coefficients between natural growing depths of seagrasses and different derivative reflectance profiles.

**Figure 9 ijerph-16-02701-f009:**
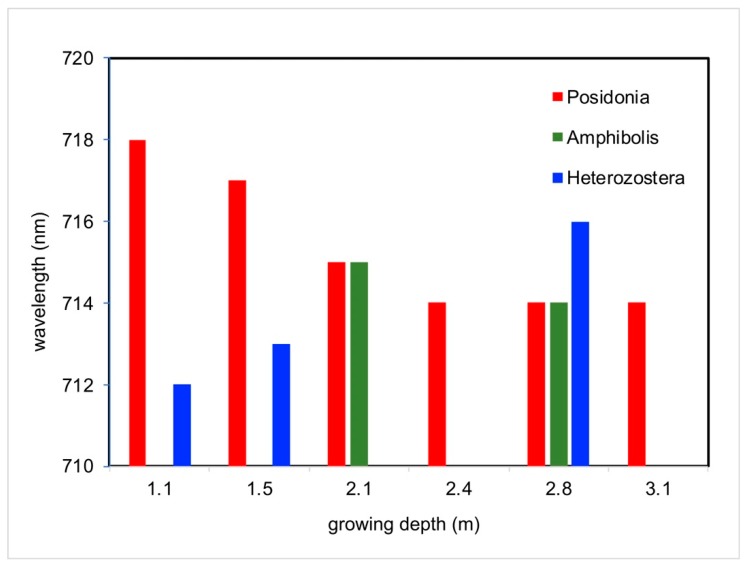
Position of “700 peak” with varying depths for *Posidonia*, *Amphibolis*, and *Heterozostera*.

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
