# Peer review of "Effects of Epiphytes and Depth on Seagrass Spectral Profiles: Case Study of Gulf St. Vincent, South Australia"

_ijerph, 2019, doi:10.3390/ijerph16152701_

Reviewer 1 Report

The manuscript entitled “Effect of epiphytes and depth on seagrass spectral profiles: Case Study of Gulf St. Vincent, South Australia” is an important study with the aim of determine the influence of epiphytes at different depths of water, and surveys of reflectance spectral signals.Results show that epiphytes significantly dampen bottom type reflectance throughout most of the visible light spectrum, excluding 670-679 nm and depth does influence reflectance, with the detection of deeper seagrasses because with increase in depth, only Heterozostera increases in exact “red edge” wavelength at which there is a rapid change in the NIR spectrum, thanks to these results the authors have helped to improve the detection of seagrass endmembers during remote sensing, thereby helping protect the natural resource of seagrasses.

The introduction is relevant but must include new references. The discussion, in the light of results and knowledge, is relevant.

Your manuscript will not be accepted unless both the technical and grammatical revisions have been made successfully.

Based on these comments, I recommend a moderate revision of analytical aspects of this manuscript before final decision about its acceptance.

Moderate comment:

Introduction:

L38-41: Rewrite this part with new references it is a very old research ... (mentioning some scientific work and innovative techniques that validate the problem of pollution that causes the decline of seagrasses ..., (Cappello et al., 2018, Parrino et al., 2019).

L38: Recent study (Cappello et al., 2018) showed the role of  metabolomics in mussel in order to assess the effects of environmental pollution.

L41: Please insert this sentence: Some authors (Parrino et al., 2019) have highlighted the effect of anthropogenic loads on the hemocyte population variation in mussels.

Reference:

(Cappello T. et al., 2018) - Baseline levels of metabolites in different tissues of mussel Mytilus galloprovincialis(Bivalvia: Mytilidae). Comparative Biochemistry and Physiology - Part D: Genomics and Proteomics. Volume 26, Pages 32-39.

(Parrino V. et al., 2019) - Flow cytometry and micro-Raman spectroscopy: Identification of hemocyte populations in the mussel Mytilus galloprovincialis(Bivalvia: Mytilidae) from Faro Lake and Tyrrhenian Sea (Sicily, Italy). Fish and Shellfish Immunology, vol. 87, p. 1-8.

Materials and methods:

L136 - 2.2. Field Data Collection

How was the procedure for collecting algae areas covered? There are previous researches ... this aspect is not clear.

L144 - Were all the algae sampled clinically healthy?  

Please describe the absence of injury.

Statistical:

The F values for ANOVA statistical analysis used are appropriate.

Results and Discussion:

The authors should be reduce this part, please the results shown in figures.

However, even the abiotic conditions of water should not be neglected.

Conclusion:

Explain how this research should help prevent the anthropogenic load produced by anthropogenic pollution, so that monitoring of marine vegetation could increase and protect marine environments.

Reviewer 2 Report

This manuscript describes the attempt to calibrate remote sensing monitoring of seagrass beds. As so, it approaches the roles of (1) seagrass species, (2) depth influence, and (3) epibiont cover on resulting remote sensing information. The manuscript is interesting at the scale of "landscape ecology" for monitoring of seagrass beds worldwide, although seagrass species vary geographically, and water conditions vary not only geographically, but also seasonally. Epibiont load may also change both seasonally and geographically. 

I am not a specialist in remote sensing but the methods and results seem sound and relatively well presented (the quality of some graphs and pictures could be improved, see comments below).

Comments and minor edits:

Introduction

L. 33 - "Seagrasses are underwater angiosperms (flowering plants) that..."

L. 34 - Not all seagrass beds are optically clear. Seagrass beds exist both in river mouths and  in coral reefs (e.g., Abrolhos bank) that are not particularly optically clear most of the time, making remote sensing assessments more difficult.

L. 36-37 - Reptiles such as some sea turtle species also feed on seagrass beds.

L. 62 - "grow on the surface of surface" remove "of surface". 

L. 85 on - "algal epiphytes" refers to algae on seagrasses, or to algae on any other basibiont organism? I particularly prefer the term "epibiont" in what it doesn't suggest the nature of the organism growing on the surface of the basibiont. "Epiphyte" sometimes refers to plants growing on animals, sometimes to animals growing on plants, which generates a lot of confusion in readers. For more clarification on these definitions I suggest reading Taylor & Wilson (2003).

L. 85 - epipfaunal: did you mean "epifaunal"?

L. 104 - "epiphytes can have been shown" remove "can"

References cited:

TAYLOR,  P.D.  &  WILSON,  M.A.  2003.  Palaeoecology  and  evolution  of  marine  hard substrate communities. Earth-Science Reviews 62: 1-103

Reviewer 3 Report

Please add further details about the cosine corrector sensor (Materials and methods section).

Please add coordinates (Figure 1).

Author Response

Please check the revised version.